# Always with the Best Intentions? Interrogating the Use of Sustainable Building Assessment Systems in Developing Countries: Kenya

**Faith Ng'eno Chelang'at** [1] and **Ranald Lawrence** [2,*]

1   Department of Architecture, Jomo Kenyatta University of Agriculture and Technology, P.O. Box 62000, Nairobi 00200, Kenya; archfaithngeno@gmail.com
2   The Liverpool School of Architecture, University of Liverpool, 25 Abercromby Square, Liverpool L69 7ZN, UK
*   Correspondence: ranald.lawrence@liverpool.ac.uk

**Abstract:** Assessment methodologies such as BREEAM and LEED allocate points based on prescribed interventions that promote design features or strategies considered to be more sustainable than others. A focus on accumulating numerical scores, however, often fails to address pertinent contextual issues, particularly within developing countries. This paper examines the use of four assessment systems in Kenya—two international systems, LEED-US and Green Star SA-Kenya; and two locally developed systems, Green Mark Kenya and the Safari Green Building Index. The paper compares the relative weighting of different categories under each system, and assesses their appropriateness to a Kenyan context, with reference to the suitability of active technology versus passive design approaches. The paper examines selected examples of 'green' buildings in Nairobi, reflecting on the influence of different methods of assessment on the adopted design approaches. The paper argues that international rating systems, such as LEED, often focus on a Western construct of sustainability featuring a systematic bias towards global rather than local perspectives, with an emphasis on physical environmental factors. In pursuit of objectivity, the measurement of non-contextual parameters untailored to local circumstances (e.g., energy performance) is prioritised at the expense of those contingent on local conditions or climate.

**Keywords:** sustainability assessment; rating system; assessment method; building performance; sustainable development; Kenya

## 1. Introduction

Assessment systems, such as BREEAM and LEED, primarily seek to define, guide, and measure environmental impact as a proxy for the sustainability or 'greenness' of buildings. While there is no consensus on terminology, this paper identifies these systems as 'Sustainable Building Assessment Systems'. These systems have distinct frameworks that assign points for different criteria classified under separate categories. Assessment systems claim to offer accepted standards that can be verified, allowing actors in the building industry to demonstrate commitment to a 'sustainable built environment' [1], or 'Leadership in Energy and Environmental Design' [2]. These systems often act as decision making tools, offering a basis for measurement of the environmental impact of buildings [3].

The introduction of rating tools began in the UK in 1990 with the development of the Building Research Establishment Environmental Assessment Method (BREEAM). LEED was first developed in 1998 by the United States Green Building Council (USGBC). Since then, many countries have developed different rating systems to measure the sustainability of buildings (Figure 1). In 2017 it was estimated that there were over 600 sustainable building rating systems in use across the globe [4].

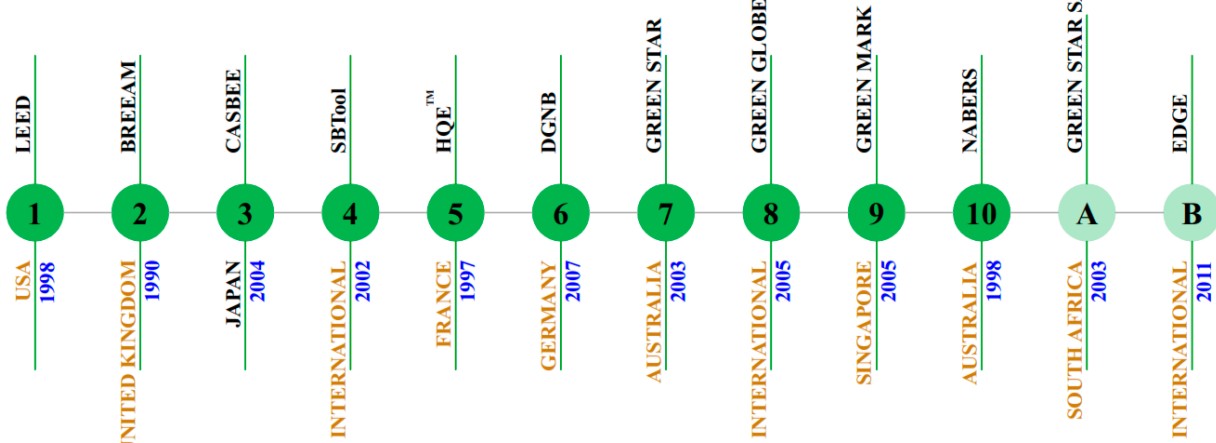

**Figure 1.** The ten dominant rating systems based on the number of buildings certified. A and B represent other rating systems that have been employed in Kenya. Author 2023. Data from Bernardi et al. [5].

While the objective of these systems is to standardise measurements, they are not static systems. In response to changing needs of the 'occupiers, investors, level of acceptance, utilisation and development of the country' [6], rating systems continuously evolve, evidenced by periodical 'upgrades'. As each country differs in attributes, such as climate, technology, culture, and economics, assessment criteria also vary. Given their voluntary nature, these systems are largely adopted by market players in an attempt to achieve a credible competitive advantage through recognition of their efforts to enhance building performance.

Assessment systems currently adopt one of two approaches. The first, and most popular, is the multi-criteria-based system which allocates points to different categories based on performance. Among these systems are BREEAM, LEED, Green Star, and Green Mark. The second and more complex of the approaches is the Life Cycle Assessment (LCA) approach, where the environmental impact of a building is 'scientifically' predicted for each of the different stages throughout the life of the building. This approach allows for assessment of the long-term impact and benefits of the building [7–9]. In developing countries, however, a lack of reliable data on building performance in use, and the embodied carbon of materials and construction, means that it is very difficult to conduct LCA in a reliable manner.

In Africa specifically, criteria-based assessment remains relatively novel. Outside South Africa, Kenya currently has the highest density of fully certified projects in sub-Saharan Africa—13 as of 2022 [10]. It is therefore imperative that built environment professionals learn lessons from the use of building assessment systems in Kenya, and consider their appropriateness for wider adoption across Africa, where a more complete understanding of the social and economic dimensions of sustainability is arguably more critical than in the West.

*Green or Sustainable?*

It is important to address the differences and implications of the terms 'green' and 'sustainable'. While the terms are often used interchangeably, they differ in meaning and methodological approach. The GBCSA defines a green building as a 'resource efficient, energy efficient and environmentally responsible building that reduce its direct and indirect impact on the environment throughout its life…' [11]. Sustainable design, on the other hand, is often considered to encompass a broader, more holistic approach. Kibert describes sustainable construction as 'creating and operating a healthy built environment based on resource efficiency and ecological design' [12]. According to Cole [13] and Dwaikat and Ali [14], sustainability considers the broader impact of buildings on the biosphere,

both at a local and global scale, throughout their life cycle. A broader definition of sustainability also considers social and economic aspects, conventionally termed 'the triple bottom line'. The *Our Common Future* report of 1987 defined sustainable development as 'development that meets the needs of the present without compromising the ability of future generations to meet their own needs' [15]. Perceived conflicts between economic, social and environmental concerns led to the concept of the 'green' economy following the Rio+20 Summit in 2012 [16,17]. and the development of 17 Sustainable Development Goals (SDGs) following the UN Sustainable Development Summit in 2015 [18]. The SDGs challenged traditional development economics by introducing the concept of inclusive growth, incorporating six separate elements: dignity, human beings, the planet, prosperity, justice, and partnership. Key to these initiatives was cooperative governance by global stakeholders to resolve conflicts between economic, social, and environmental goals [19].

Better understanding of these terminological differences casts light on competing interpretations of 'green' or 'sustainable' design. While social and economic aspects are not new considerations in design, very little has been done to integrate them into rating systems that measure performance. Living standards in developing countries remain far below developed countries. While it is important to embed environmental responsibility, it is right that there is a simultaneous focus on ensuring investment in sustainable products and services makes a worthwhile difference in these contexts.

## 2. Rating Systems in the Developing World

Guy and Moore argue that the development of assessment systems often hinges on two major assumptions; one, that environmental concerns are 'physical in nature and global in scale' and two, that 'rational science can and will provide understanding of the environment necessary to rectify environmental bads' [20]. This universalising tendency often overlooks local environmental problems, reducing assessment to the physical characteristics of a building (e.g., end energy use), failing to take into account other contextual factors (e.g., available energy resources, construction methods), or qualitative (human) aspects of sustainable design that emerge once buildings are integrated within the broader cultural and socio-economic fabric of a particular context. Zuo and Zhao highlight how rating systems prioritise a wide range of environmental issues but do not give equal consideration to social and economic aspects of sustainability [21]. Cole emphasises that the 'context within which an assessment method has been designed to operate profoundly affects the effective scope, emphasis and rigour of an assessment' [13].

While there is a growing international consensus that the 'environmental' performance of buildings is inextricably related to global climate change, economic and social issues are of higher priority in developing countries compared to developed countries [22,23], and therefore there is a qualitative contrast when assessing environmental considerations against local impediments. Ebohon and Rwelamila suggest that a significant difference between developing countries and Western countries is the lack of established local institutions in the former to influence the development of sustainable policies and behaviour in the construction industry, leading to the imposition of 'policies that are largely incongruent with the peculiarities of these economies' [24]. For example, examination of the per capita carbon footprint of the developing world shows that carbon emissions from the 46 least developed countries account for only 9% of the world's average [25]. Lower energy demand means there is more potential for energy to be sourced from sustainable sources. For example, renewable hydropower dominates energy generation in many African countries, including the Democratic Republic of Congo, Ethiopia, Malawi, Mozambique, Uganda, Zambia and Kenya [26]. Hydropower projects currently under construction will more than double power generation in East Africa in the coming decade [27]. There is, therefore, little benefit in focusing limited resources on improving building energy performance in these countries.

The dynamics of globalisation, however, have increased the tendency for rating systems to be applied interchangeably across international markets. It is important to examine,

therefore, whether a building assessed under one rating system would achieve a similar outcome when subjected to another. A study by Roderick et al. compared the energy performance of a typical building using LEED, BREEAM and Green Star [28]. The same building received a low energy rating employing BREEAM but scored highly when Green Star was applied. The same building did not meet energy certification under LEED criteria. Similarly, a study by Reed revealed that buildings that attained BREEAM Excellent, LEED Platinum or 6-star Green Star ratings would be likely to feature very different sustainable design considerations and mitigations [29].

Another study by Chen et al. analysed the passive design approaches of five green building rating tools (BREEAM, LEED, CASBEE, BEAM and GBL-ASGB) [30]. The study concluded that most green building rating tools do not accord passive design strategies appropriate weighting compared to traditional whole building energy simulation approaches. Furthermore, LEED reduces the available credits for passive design strategies, creating a bias towards mechanical ventilation. This is clearly punitive to many countries in Africa. The climates of Nairobi and Mombasa in Kenya, for example, favour passive design strategies over mechanical ventilation [31,32].

An interrogation of the institutions and agendas behind the development of these systems may be informative. LEED, for example, is administered by Green Business Certification Inc. (GBCI), founded in 2009 to expand the market for certification, and 'avoid any perceived conflicts of interest' between the ongoing development of the standard and certification as a product offered by USGBC members [33]. The GBCI's primary strategic goal is to 'evolve a global business model that is scalable, accessible and sustainable for delivering credentialing and certification programs', of which LEED is the most well-established [34]. The USGBC is dominated by a corporate membership directly offering or financing products and services to the construction sector. Of a current membership of 4421 organisations, there are 1758 architects/engineers, 590 contractors, 430 specialist consultants and 362 product manufacturers, of which 3751 are based in the US [35]. It is not surprising, therefore, that there is an in-built bias towards the specification of high-value technological solutions most effective in a US context, which also enhance the contract value of services offered by manufacturers and built environment professionals. While LEED is marketed as representing 'a transformative milestone in the built environment's alignment with a low-carbon future' [2], evidence for improvement in energy performance of LEED-certified buildings remains inconclusive [36–40]. In a survey of 106 studies of rating systems, Geng et al. concluded that there was no clear relationship between building rating and energy use [41].

*Current Research*

A recent trend has been to propose entirely novel rating systems, either for previously overlooked categories of buildings [42], or different geographical contexts [43]. Other papers have explored the possible integration of assessment methodologies [44]. A recent study examined and categorised 101 assessment systems based on the method(s) employed, the different aspects of sustainability included, and the type(s) and stage(s) of project covered [45]. The study concluded that most systems focused on energy efficiency and indoor environment quality at the expense of social and economic aspects.

Díaz-López et al. highlighted that assessment systems are most prevalent in Europe and North America, followed by Asia [45]. 70% of assessment systems are developed by Green Building Councils formed of private companies and other organisations representing built environment professionals, overseen by the World Green Building Council. Very few systems are designed or developed in developing countries, particularly Africa, due to the economic costs involved. Song et al. highlight the Western origins of rating systems, reflecting a 'standardised expression of the evolution of green building concepts and a roadmap for future directions of both market and academia' [46].

The majority of research papers examining the use of rating systems in international contexts address the applicability of LEED, focusing on well-rehearsed problems with the

assessment of energy and material sourcing [47]. Discussing the dominance of LEED over the previous two decades, Song et al. argue that its use in over 160 countries demonstrates its 'flexibility', while simultaneously acknowledging that it evaluates design related criteria differently to systems 'customised and employed in local contexts', such as Green Mark [46].

Existing research exploring the adaptation or development of new standards largely focuses on the Middle East, with a particular emphasis on Indoor Environmental Quality (IEQ) and cultural concerns [48]. One of the few papers to consider sub-Saharan Africa examines the development of a proposed new rating system for Nigeria, based on interviews with experts in the region. The paper concludes that existing rating systems disproportionately prioritise IEQ and energy criteria at the expense of social and economic factors [49].

Reflecting on recent developments, Felicioni et al. argue that a focus on sustainability has meant that the concept of resilience has been overlooked [50]. The authors claim that 'synergies' between the concepts provide opportunities for integration in building design to mitigate the worst aspects of climate change. Their study focuses on Europe, citing moves towards mandatory certification in the UK, Germany and Italy as positive steps towards improving the overall quality of the built environment, without considering the appropriateness of these policies in a developing world context, where resilience to climate change is arguably more critical.

Another recent study has highlighted inconsistencies and discrepancies that exist between rating systems, and the relative absence of factors reflecting social aspects of sustainability [51]. Other authors have advocated for the inclusion of specific categories that recognise the social dimension of sustainability—in particular social justice, equity, education and culture [52].

It is evident that the assessment of building performance can vary significantly based on the methodology that is applied and the context in which it is applied. This is as a result of differences in the assessment method and performance criteria. Often, assessment reflects an attempt to outline the benefits of a building that has been through a rating process, e.g., in terms of carbon footprint or occupant wellbeing, with a base-line alternative. It is, however, very difficult to achieve an objective assessment as individual buildings are highly context dependent. Comfort, for example, has been described as a 'highly negotiable socio-cultural construct' [53], yet standards are often adopted universally, leading to the ubiquitous use of mechanical systems to achieve narrowly defined thermal conditions irrespective of context.

Building codes and standards also differ between countries. Not only do some have far more stringent regulations than others, but regulatory priorities also differ. While there have been attempts to develop rating systems that can be adapted to local contexts in terms of physical parameters, such as water and energy, adaptation of assessment systems to local social, economic, and technological advancement levels remains challenging [54]. Not only do international systems fall short in addressing local issues, of greater concern is their potential to stifle local solutions that may be more appropriate within a developing world context. As of present, no existing body of research focuses on the overarching structural injustices presented by the building assessment methodology paradigm to developing world countries.

Contrary to much of the existing literature, rating systems do not primarily evolve in response to scientific advance, but in response to political and market trends. Businesses directly involved or funding construction face pressure to address their environmental impact. Rating systems provide a convenient rubric through which the built environment industry can demonstrate engagement and improvement of performance within normative boundaries established to ensure that sustainability does not disrupt established business models, while providing an opportunity to 'add value' for their clients. An example of this is the way in which energy and carbon use is often assessed, based on simulation models that bear little relation to how buildings operate in the real world. Methodologies based on

a process of rational scientific evolution would have responded to the growing scientific consensus around this issue, known as 'the performance gap' [55].

Similarly, but less recognised, is the impact of these methodologies in contexts outside those for which they were originally designed and developed. The promotion of assessment systems in the developing world represents a secondary business opportunity, pursued in response to growing global political consensus on the impacts of climate change and the need for all countries to take mitigative action. Assessment is offered as a service to improve the performance of prestige buildings, typically designed to offer what is perceived as a Western level of comfort/specification (e.g., more 'efficient' technology, 'higher-spec' materials or components, indoor and outdoor planting systems, bicycle shelters, etc.) in contexts where this is not the norm.

### 3. Building Regulations and Rating Systems in Kenya

Building regulations in many African countries are often based on colonial-era codes which mandate minimum requirements for construction. In Kenya, draft National Building Regulations to replace the 1968 Building Code were published in 2015, covering appropriate use of materials and technology, maintenance, accountability, and professional services. The regulations do not mandate minimum standards for energy or fabric performance but prioritise passive measures. Part N recommends that all projects 'should make provision for adequate natural lighting, natural cooling and natural ventilation', including for example 'appropriate choice of construction materials together with sun shading and other controls of heat gain or loss', and 'the natural removal of any heat gains' utilising 'various forms of natural ventilation' [56]. The Regulations have yet to be enacted.

The first three buildings to be certified using a rating system in Kenya, all completed in 2015, employed internationally established versions of LEED. A further eight buildings have since been LEED certified. Intended from the outset to lead the market, LEED has undergone multiple revisions since its introduction [33]. LEED 2.0, released in 2000, expanded its scope to include existing buildings and interiors. LEED 3.0, released in 2009, incorporated 4 'regional priority credits' out of a total of 110, in an effort to demonstrate that diverse geographic contexts could be accommodated by the standard. LEED 4.0. released in 2013, focused on performance-based metrics, including lifecycle assessment, with an increased emphasis on materials and resources. LEED version 4.1, released in 2019, incorporated changes to streamline the certification process and encourage higher levels of sustainability performance [57]. Unlike BREEAM, LEED was originally designed to reward energy saving measures rather than a direct reduction in energy use. The intention was to prevent a situation where 'energy hungry building with "clean" grids would receive a higher rating than energy efficient buildings with "dirty" grids' [33]. An examination of LEED criteria demonstrates that it is more straightforward to evidence energy efficiency measures through technology-based systems rather than passive design measures.

Green Star SA-Kenya was launched in 2017 by the Green Building Council of South Africa (GBCSA), under license from the Green Building Council of Australia (GBCA). According to GBCSA, Green Star 'is a natural touch point for green building movements and councils in other parts of Africa', once identified adaptations have been made to take account of local context [58]. Two buildings have been certified employing the system.

Green Mark Kenya was launched by the Green Africa Foundation in 2018, with support from the Government of Kenya, the United States Agency for International Development (USAID) and the United Nations Development Programme (UNDP). Commenting that 'internationally devised rating systems have been tailored to suit the building industry of the country where they were developed', the authors of Green Mark argue that 'a Kenyan Standard for Green Buildings is critical if the country is to transition to a truly low-carbon, climate-resilient and sustainable development pathway' [59].

Finally, the Safari Green Building Index was launched in 2021 by the Architectural Association of Kenya, aimed at promoting 'efficient bio-climatic construction practices and

optimisation of locally available materials' specifically for climate zones in Kenya and East Africa [60].

Despite the recent proliferation of rating systems in the country, only 13 completed projects in Kenya are fully certified as of 2022, with a further 20 in progress [10,61]. All of these projects are located in or around Nairobi except for one located in Mombasa. Eleven of the fully certified projects employed LEED, and two Green Star SA-Kenya (see Table 1). Currently no projects have been fully certified using Green Mark Kenya or the Safari Green Building Index. A further four projects have been certified by EDGE; however, the EDGE methodology is not comparable in scope. EDGE is promoted by the USGBC and World Bank as a tool to 'scale up resource-efficient buildings in a fast, easy, and affordable way' in emerging economies. It compares the energy and water use of a proposed design against a typical local building [62].

**Table 1.** Projects certified in Kenya by rating system. Source Author 2023.

| | Building/Development | Certification | Points | Year |
|---|---|---|---|---|
| **LEED Certified Developments** | | | | |
| 01 | Eaton Place | Certified | 47 | 2015 |
| 02 | World Bank Group | Gold | 67 | 2015 |
| 03 | Citibank Gigiri Branch | Gold | 64 | 2015 |
| 04 | Wrigleys' Nairobi Confection | Gold | 61 | 2018 |
| 05 | Campus Diplomatique Francais Nairobi | Gold | 61 | 2019 |
| 06 | Lumen Square | Silver | 61 | 2022 |
| 07 | ICRC's Nairobi Regional Delegation | Gold | 68 | 2022 |
| 08 | Microsoft Office Nairobi | Platinum | 83 | 2022 |
| 09 | Microsoft Office Nairobi Phase 02 | Gold | 72 | 2022 |
| 10 | Vienna Court | Gold | 61 | 2023 |
| 11 | Capital M Apartments | Silver | 52 | 2023 |
| **Green Star SA-Kenya** | | | | |
| 01 | Garden City Village Phase 01 | Certified | 46 | 2018 |
| 02 | Dunhill Towers | Certified | 60 | 2018 |
| **EDGE Certified Developments** | | | | |
| 01 | Britam Towers | Certified | N/A | 2018 |
| 02 | The Promenade | Certified | N/A | 2020 |
| 03 | Riverside Cube | Certified | N/A | 2021 |
| 04 | Purple Tower | Certified | N/A | 2021 |

While a lack of certified projects in Kenya makes a comparison between rating systems more speculative, it provides an opportunity to reflect on the design of these systems while they are still in their early adoption phase, informing their future use in Kenya, but also the development of analogous tools in other African countries.

## 4. Methods

This paper explores the conflicts surrounding the use of rating systems in Kenya through a comparison of the four primary systems currently promoted in the Kenyan construction market. Two are international systems, LEED-US, and Green Star SA-Kenya; and two are locally developed tools still in their draft stages, Green Mark Kenya and the Safari Green Building Index (Table 2). By comparing the latest versions of their assessment

manuals, it is possible to establish commonalities and divergences in their approaches and examine their respective priorities. The rating systems are compared by analysis of the scoring criteria (Appendix A) and harmonisation of their different scoring categories, so that the weighting afforded to different aspects by each assessment system can be compared (Table 3).

**Table 2.** Aggregation of rating systems in this study. Source Author 2023.

| LEED 4.0 BD+C (New Construction) | Green Star SA-Kenya | Green Mark Kenya | Safari Green Building Index |
|---|---|---|---|
| Platinum (80+ points), | Six Star (75+%) | Diamond (91–100 points) | Platinum or Five Star (80+ points) |
| Gold (60–79 points) | Five Star (60–74%) | Platinum (85–90 points) | Gold or Four Star (70–79 points) |
| Silver (50–59 points) | Four Star (45–59%) | Gold (75–84 points) | Silver or Three star (60–69 points) |
| Certified (40–49 points) | | Silver (65–74 points) | Bronze or two star (50–59 points) |
| | | Bronze (50–64 points) | |

**Table 3.** Comparison of the rating systems categories and weighting. Source: Author 2023.

| Category | LEED 4.0 BD+C | % | Green Star SA-Kenya | % | Green Mark Kenya | % | Safari Green Building Index | % |
|---|---|---|---|---|---|---|---|---|
| Operations | Integrative process | 1 | Management/emissions | 17 | Maintenance and management | 10 | Prerequisite requirements | 0 |
| Transport | Location and transport | 15 | Transport | 9 | (Under sustainable site) | | - | - |
| Site/ecology | Sustainable sites | 9 | Land use and ecology | 7 | Sustainable site, planning development and management | 15 | General description/recommendation | 5 |
| Water efficiency | Water efficiency | 10 | Water | 14 | Water efficiency | 10 | Water supply and drainage | 10 |
| Energy efficiency | Energy and atmosphere | 30 | Energy | 25 | Energy efficiency | 20 | Energy efficiency | 10 |
| Materials And resources | Materials and resources | 12 | Materials | 13 | Sustainable materials and technology | 10 | Resource efficiency | 20 |
| Indoor Environmental Quality | Indoor Environmental Quality | 15 | Indoor Environmental Quality | 15 | Indoor Environmental Quality | 30 | Passive design strategies | 45 |
| Innovation | - | 5 | - | - | Innovation | 5 | Innovation | 5 |
| Other | - | 3 | - | - | - | - | Noise and acoustic design | 5 |

For detailed comparative purposes, harmonisation of categories is carried out in order to create a system of common categories. Some original categories are relocated to fit the developed categories. The harmonisation process involved categorising 'passive design' from the Safari Green Building Index under 'Indoor Environmental Quality', and 'acoustic design' under 'Other'. 'Water supply and drainage' was extrapolated from 'resource efficiency'. Similarly, the 'transport' metric in the Green Mark tool was extrapolated from the 'sustainable site, planning development and management' category. Finally, the 'management' category in Green Star and Green Mark, 'integrative process' in LEED, and 'prerequisite requirements' in Safari Green Building Index were grouped together under 'Operations', and the 'emissions' category in Green Star was combined with 'management'.

In order to compare points from each category, a normalisation process scaled all the categories to a common whole (Table 4), with all scoring criteria converted to percentages. The 'Innovation' and 'Other' categories were eliminated.

**Table 4.** Harmonised weighting of categories for different rating tools. The most significant categories are highlighted in bold. Source: Author 2023.

| Category | LEED 4.0 BD+C | Green Star SA-Kenya | Green Mark Kenya | Safari Green Building Index |
|---|---|---|---|---|
| Site/ecology | 10% | 7% | 15% | 5% |
| Energy efficiency | **33%** | **25%** | 20% | 10% |
| Water efficiency | 11% | 14% | 10% | 10% |
| Materials and resources | 13% | 13% | 10% | 20% |
| Indoor environmental quality | 16% | 15% | **30%** | **50%** |
| Transport | 16% | 9% | 9% | 0% |
| Operations | 1% | 9% | 10% | 0% |

## 5. Results: Comparative Analysis

It is apparent that, in all four rating systems currently employed in Kenya, categorisation of criteria reflects a wide range of different design and construction practices and processes. These categories may differ in terminology, but they generally feature similar objectives, permitting comparisons between categories in different systems to be established. Differences in the weightings of each category provide evidence for differences in priorities between systems (see Table 4).

The 'Energy efficiency' category, for example, shows significant differences. Comparing the harmonised category weightings in LEED and Green Star SA-Kenya—the international systems—the energy categories are assigned the greatest importance, accounting for 33% of the assessment score in LEED and 25% in Green Star SA-Kenya, whereas in Green Mark Kenya and Safari Green Building Index, Energy efficiency is assigned 20% and 10%, respectively. On the other hand, 'Indoor environmental quality' is given the greatest emphasis in both Green Mark Kenya and the Safari Green Building Index, accounting for 50% and 30%, respectively.

Analysis of the Energy efficiency category reveals that LEED is biased towards active strategies for minimising energy use, e.g., commissioning, optimisation and 'Demand response' of building services; maintenance of equipment (e.g., 'Refrigerant management'); and 'Renewable energy production' (Appendix A). These strategies are not only capital intensive but also high in embodied energy, as mechanical services are neither produced nor assembled in Kenya.

Green Star SA-Kenya does not prioritise the efficiency and operation of building services in the same way but focuses on reduction of demand and monitoring of energy, with lighting being a key consideration. This could be due to its adaptation from Green Star SA. South Africa has suffered an energy supply crisis that has resulted in a series of disruptive outages and load shedding since 2008, and from this point there has been a gap between electricity demand and supply.

Green Mark Kenya and Safari both include criteria for 'Energy efficiency of equipment, appliances, fittings' and 'Renewable energy', with Green Mark also including criteria for Optimisation and Efficiency of building services.

It is worth noting that the energy mix in the countries where the systems originate differs significantly. Approximately 70% of Kenya's energy is generated from renewable sources [63]. This is significantly higher when compared to South Africa, where energy generation is dominated by coal power at approximately 89%, with renewable energy constituting about 3% [63]. In the US, renewable energy constitutes approximately 21% of

the energy mix [64]. Reducing energy use is, therefore, crucial to cutting carbon emissions and mitigating the environmental impact of buildings in these countries. South Africa, for example, is responsible for the largest share of greenhouse gases in Africa, with carbon emissions not projected to peak until 2035 [65].

The respective climates of cities in Kenya, South Africa, Australia and the US provide an indication of the requirement for heating and cooling in different regions at different times of the year (Figure 2). The climates of Nairobi and Mombasa are characterised by warmer temperatures throughout the year, for longer periods of the year, when compared to, e.g., Johannesburg, South Africa, or New York in the US. Compared with a hot dry climate like Phoenix, US, the range of temperatures in Nairobi and Mombasa is closer to the human comfort zone across the annual cycle.

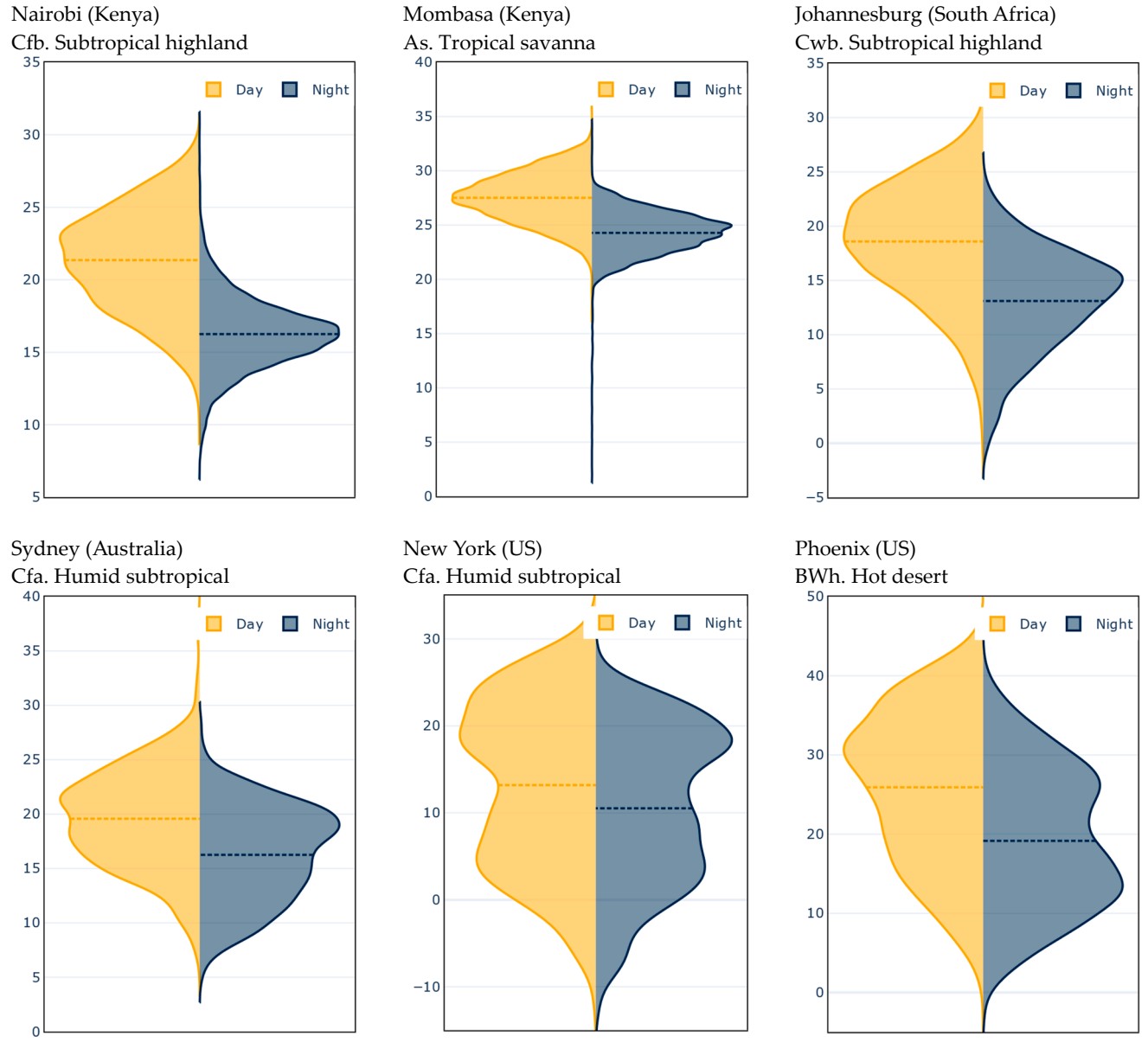

**Figure 2.** Probability of day and night dry bulb temperatures (°C) at selected cities, with Köppen–Geiger climate classification. Source: Betti et al. [66].

In Nairobi and Mombasa, passive design strategies can therefore be utilised to significantly reduce the energy demand for cooling and achieve thermal comfort. At equatorial latitudes, solar shading becomes an imperative part of design as a way to reduce the cooling load within a building. Similarly, the heating load can be avoided through passive design

strategies, such as cross-ventilation. A focus on passive strategies favours the economic context as they tend to be less capital intensive, especially when compared to the potential energy savings. Despite this, only Safari offers credits specifically for an alternative passive approach, e.g., taking into consideration 'Building Orientation' 'Building form and shape', 'Openings', 'Passive Heating or Passive Cooling', and 'Natural Ventilation'. In contrast, LEED prioritises response to extreme weather conditions, which increases a building's energy load for both cooling and heating [30].

The 'Indoor Environmental Quality' category also shows significant differences. This category assesses thermal, visual, and acoustic comfort as well as ventilation, air quality and pollution. Whereas international systems tend to prioritise Energy efficiency, Kenya's locally developed systems prioritise IEQ. Safari allocated about half of its credits to IEQ while Green Mark Kenya allocated approximately one third. This is in contrast with LEED and Green Star SA-Kenya, which allocate 15% and 16% respectively to IEQ. Furthermore, within this category, greater emphasis is placed on different criteria. LEED for instance emphasises air quality, while Safari focuses on passive solar control strategies. The difference in climate in the three countries where these systems originate (with Green Star SA heavily borrowing from Green Star Australia) perhaps explains these differences.

The 'Water efficiency' category considers the harvesting, consumption, reduction in use, monitoring, storage, and recycling of grey water. Analysis of all four assessment systems reveals a relatively similar weighting allocation to water efficiency. However, the local assessment systems' weightings are lower than the international systems'. An analysis of these contexts would suggest that water requirements and challenges differ across the different countries. Kenya is a water scarce country, especially in comparison to South Africa and the US [67]. Conversely, one of Kenya's greatest environmental challenges is flooding. As a result of climate change and rapid urbanisation, not only is there a strain in available water resources, but there has been a significant increase in poorly planned building density, coupled with poor urban drainage systems, that has resulted in a flood risk crisis in Kenya's urban environment [68]. This suggests, therefore, that the design of an assessment system appropriate for Kenya would allocate more weight and a greater number of credits or points to buildings that mitigate these challenges.

The 'Materials and resources' category considers relatively similar criteria, including the management processes involved in the choice, sourcing, use, storage, and recycling of building materials throughout the building process. The toxicity of materials as well as waste management are considered in all four systems. The weighting in this category shows marginal differences in three of the systems, with both Green Star SA-Kenya and LEED allocating 13%, and Green Mark Kenya 10%. Safari allocated 20% to this category, making it its second most significant category after IEQ.

Finally, the 'Transport' category looks at access to mass transit systems, the use of non-motorised transport and energy efficient transport. This is reflected in all assessment systems except Safari, though with different weightings. LEED prioritises transport when compared to the other assessment systems, allocating 16%, which is its second highest category, whereas Green Star and Green Mark both allocate 9% to transport. Interestingly, 53% of $CO_2$ emissions in Kenya are produced by transport. This is significantly high compared to 12% in South Africa and 33% in the US [69]. Notwithstanding the rapid surge of vehicle ownership in Kenya, the use of electric cars is still in its infancy, and therefore the infrastructure to support this technological advancement is yet to be developed or prioritised. Furthermore, despite plans to develop a Mass Rapid Transit System, urban areas in Kenya currently rely on individual transport (vehicles) and privately owned buses, commonly referred to as *matatus*.

*Examples in Nairobi*

The first three buildings to attain LEED certification in Nairobi are modern office buildings, including financial premises for Citibank Gigiri Branch and World Bank Group (both certified for Interior Design and Construction), and Eaton Place, a business centre

for commercial rent (certified for Building Design and Construction, known as 'Core and Shell'). A comparison of the former two buildings that attained Gold suggests that the most difficult categories to attain points were Indoor Environmental Quality at 47% and 41%, respectively, and Materials and Resources, where projects attained 0% and 35%, respectively (Table 5).

**Table 5.** Comparison of LEED (ID+C 3.0) certified buildings in Nairobi. Source: Author 2023.

|  | Citibank Gigiri Branch | | World Bank Group | |
| --- | --- | --- | --- | --- |
| **Classification Sort** | LEED ID+C 3.0 2009 | | LEED ID+C 3.0 2009 | |
|  | **Total** | **Attained** | **Total** | **Attained** |
| Categories |  |  |  |  |
| Sustainable Site | 21 | 19 | 21 | 18 |
| Water Efficiency | 11 | 6 | 11 | 11 |
| Energy and Atmosphere | 37 | 22 | 37 | 20 |
| Materials and Resources | 14 | 0 | 14 | 5 |
| Indoor Environmental Quality | 17 | 8 | 17 | 7 |
| Innovation | 6 | 5 | 6 | 2 |
| Regional Priority | 4 | 4 | 4 | 4 |
| Total Score | 110 | 64 | 110 | 67 |
| Certification Attained | Gold | | Gold | |

These results are similar to a study examining LEED (ID+C 3.0) certified buildings in China and the U.S., which found that the median number of points attained for Indoor Environmental Quality was 50% in China and 53% in the U.S., and 29% in China and 32% in the U.S. for Materials and Resources [70]. Both categories also proved most challenging for Eaton Place, which attained 5% and 15%, respectively (Table 6). Preliminary analysis suggests, however, that these two categories should be easier to achieve within a Kenyan context. While Indoor Environmental Quality has been identified as problematic in developed world contexts within the design constraints imposed by LEED, particularly in terms of individual user control of their environment [71,72], the benign nature of the Kenyan climate should lead to improved performance in that context (see Figure 2). Similarly, while construction materials are typically locally sourced, problems with auditing mean that more expensive materials are often required to be specified from further afield [73].

In a separate paper the authors examined the environmental design in case studies of 'green' buildings in Nairobi [74]. The Strathmore Business School was to be the first new-build LEED certified building in Kenya before certification was abandoned. Pursuing LEED certification added to the budget, primarily due to an inadequate understanding of the process, the cost of learning, procurement, and the import of technology, e.g., photovoltaic glass and louvres specified to attain points for renewable energy. A tender sum of KSh. 780 million was approved in 2008, equating to approximately KSh. 78,000/m$^2$ or US $1100/m$^2$, compared with an average construction cost for school buildings in Kenya of US $170/m$^2$ [75]. One consequence of this was that the atrium, originally designed with a retractable roof, was simplified to feature fixed glazing only (Figure 3), which significantly affected the intended indoor environmental quality performance of the space. Prioritising LEED significantly increased the budget in order that renewable energy could be generated on-site (despite the Kenyan grid being largely powered by renewable energy), and an inappropriate modification to the central space of the building limited the ability of the building occupants to adapt their environment to the changing weather and seasons, impacting indoor environmental quality.

**Table 6.** LEED (BD+C 3.0) certified buildings in Nairobi. Source: Author 2023.

| | Eaton Place | |
|---|---|---|
| **Classification Sort** | **LEED BD+C 3.0 2009** | |
| | **TOTAL** | **ATTAINED** |
| Categories | | |
| Sustainable Site | 28 | 14 |
| Water Efficiency | 10 | 6 |
| Energy and Atmosphere | 37 | 16 |
| Materials and Resources | 13 | 2 |
| Indoor Environmental Quality | 12 | 1 |
| Innovation | 6 | 4 |
| Regional Priority | 4 | 4 |
| Total Score | 110 | 47 |
| Certification Attained | | Certified |

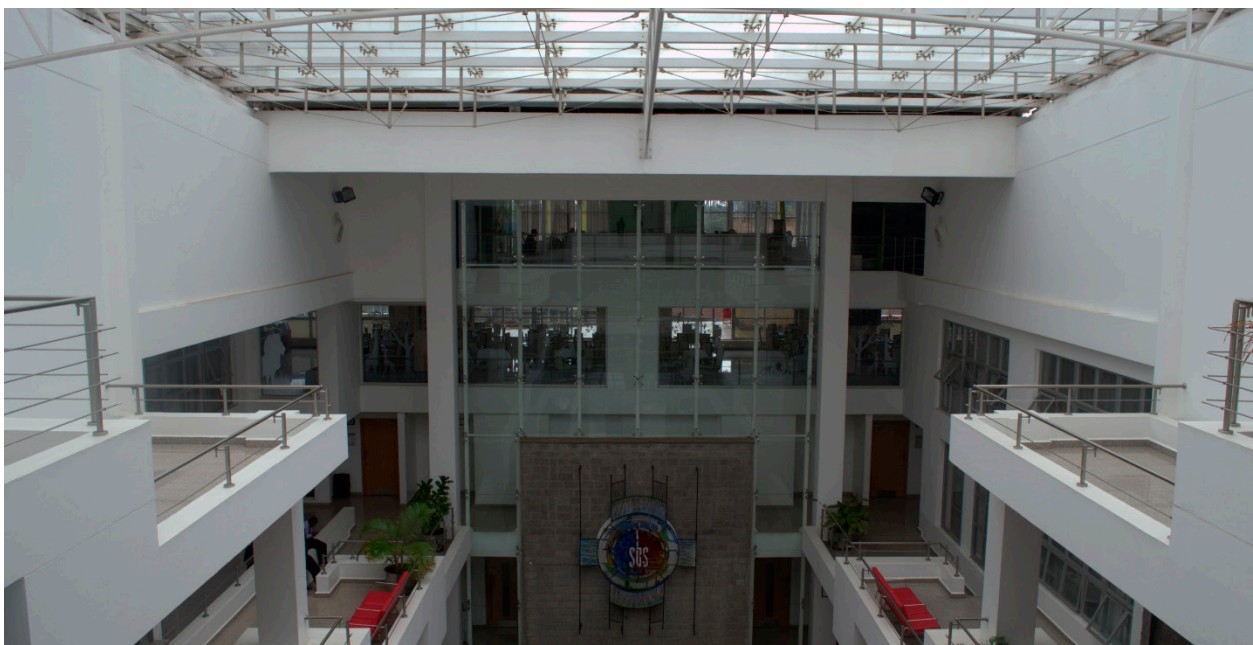

**Figure 3.** Strathmore Business School atrium. Author 2023.

Similarly, another project with significant design input from a multinational engineering firm based in the UK (Kenya Commercial Bank Towers) was designed by its local architect with minimal requirements for air conditioning, relying on passive ventilation, solar shading and thermal mass to regulate the diurnal and seasonal variation in indoor temperatures. However, the engineers recommended that key spaces be sealed with air conditioning installed to ensure comfort on the hottest days of the year, a key criteria for LEED assessment. Post-occupancy surveys reveal that the building is often perceived to be too cold, requiring the use of portable heaters. The design modifications led to a significant increase in construction cost of KSh 800 million, on top of an initial tender price of KSh 1.8 billion, as well as increased operational energy expenditure for both heating and cooling.

These examples show how a focus on the technological solutions and innovation encouraged by LEED could lead to design solutions that are economically unsustainable,

where comfort and user experience is sacrificed by misguided attempts to improve simulated rather than real-world energy performance.

## 6. Discussion

Previous studies have found that the cost of achieving certified green buildings is considered prohibitive, especially in developing countries [76–80]. It is, therefore, imperative that the most appropriate rating system is selected so that any potential benefits outweigh the costs of assessment.

This research found that criteria vary significantly between rating systems in use in Kenya (Appendix A). LEED, Green Star SA-Kenya, and Green Mark Kenya feature a comparatively higher number of assessment criteria compared with Safari Green Building Index. The research attempted to analyse and classify the criteria in the four assessment systems under the key pillars of sustainable design: environmental, social, and economic (Table 7). Some criteria overlap the Environmental and Social pillars (e.g., Bicycle Facilities)—these have been allocated to the Social category. An additional 'Administrative' classification includes criteria such as employing accredited professionals or regional priority.

**Table 7.** Criteria weighting under key pillars of sustainability. Source: Author 2023.

| | LEED 4.0 BD+C | | Green Star SA-Kenya | | Green Mark (Kenya) | | Safari Green Building Index | |
|---|---|---|---|---|---|---|---|---|
| | No. | % | No. | % | No. | % | No. | % |
| Environmental | 27 | 51 | 36 | 54 | 29 | 49 | 19 | 66 |
| Social | 11 | 21 | 12 | 18 | 16 | 27 | 4 | 14 |
| Economic | 0 | 0 | 1 | 1 | 6 | 10 | 1 | 3 |
| Admin | 15 | 28 | 18 | 27 | 8 | 14 | 5 | 17 |
| Total | 53 | 100 | 67 | 100 | 59 | 100 | 29 | 100 |

From the analysis it is evident that all four assessment systems distinctly prioritise environmental considerations, reflecting 49–66% of total criteria. Less consideration is given to social criteria despite the popularity they are gaining in sustainable design discourse. Green Mark Kenya allocated the highest proposition at approximately 27% of total criteria, LEED allocated approximately 21%, Green Star SA-Kenya allocated 18%, and Safari allocated 14%. Economic considerations, however, are still barely considered in all systems. Green Mark allocated 10%, Safari allocated 3%, Green Star allocated 1% and LEED allocated 0%. It is clear, however, that green building design and assessment should extend beyond merely minimising environmental impact; it should equally prioritise the long-term health and well-being of occupants, along with the potential for cost savings. This holds greater significance for developing nations, as numerous studies indicate that the economic and social aspects of sustainable design play a more crucial role in these countries [23,78–80].

*Response to Climate Context*

The interconnections between the building envelope, mechanical systems, passive systems, and the building context are significantly influenced by climatic conditions. It is crucial to take into account climatic conditions not only when evaluating energy efficiency but also when assessing thermal comfort, indoor air quality, and daylighting levels. Energy performance, however, as outlined in the Western assessment systems, predominantly relies on energy modelling simulating the performance of active systems. Assessing potential energy performance and waste in buildings poses challenges when utilising passive systems as computational models are largely based on the predicted performance of technological interventions. Furthermore, the challenges associated with energy modelling encompass issues related to data availability, the intricate nature of modelling tools and technology, as well as the considerable time and cost involved in computation [81,82]. These challenges

are further compounded in developing countries where the availability of resources for achieving sustainable buildings shows notable variations compared to the Western world.

In a Kenyan context, research suggests that passive design approaches are the most appropriate to ensure economic sustainability and occupant comfort. The adaptive thermal comfort model is a more accurate predictor of comfort in mixed-mode as well as naturally ventilated buildings in subtropical climates such as Nairobi, where air conditioning is unnecessary except in unusual circumstances [83]. Analysing the seasonal variation in temperature and humidity in Nairobi, Rabah and Mito conclude that mean day time temperatures fall within the comfort range of 21–28 °C in warmer months (January–April and October), and 20–25 °C in cooler months (May–September and November–December), and do not cause undue thermal stress [31]. They suggest that encouraging air movement is a desirable design strategy, with additional thermal capacity aiding comfort in warmer months. As the examples discussed here illustrate, however, international standards prioritise predominantly technological interventions. Furthermore, assessment systems are still not adequately tailored to local climate. For instance, despite the four bonus points introduced in the regional priority category in LEED, structures with similar design and construction methods situated in different climatic regions will attain similar scores despite their performance differing substantially [84].

## 7. Conclusions

This paper urgently reinforces the need to prioritise assessment criteria to address local conditions. The analysis demonstrates how different contexts can influence the development and application of assessment systems, highlighting the importance of contextual diversity and its overall influence on sustainable design. The locally developed systems analysed in this paper show clear differences in their prioritisation of criteria when compared to international systems. There is an apparent attempt to contextualise the systems, but there remains much room for improvement.

Prescriptive international solutions are representative of a technocratic Western construct of the environmental problem, from a global perspective. The major global environmental challenge is currently climate change, the main mitigation of which is the reduction of carbon emissions across the globe. For buildings, this translates to the reduction and efficient use of energy, and as a result building performance assessment prioritises energy efficiency, as evidenced in the assessment tools studied. These models may be appropriate for Western countries; however, for countries like Kenya, whose energy largely originates from sustainable sources, the focus on energy as a main performance indicator is not appropriate. While Kenya faces the consequences of climate change, reducing carbon emissions does not represent an immediate priority in a country that accounts for only 0.13% of the global total. Countries like Kenya should commit to global environmental responsibility; however, the characterisation of the local environmental challenge differs significantly from the Western and global characterisation of the problem. Framing the actual local environmental challenge will ensure that mitigative measures are relevant, appropriate, and effective.

The analysis suggests that a more appropriate focus for sustainable design in developing countries like Kenya would be social and economic factors as outlined by the UN Sustainable Development Goals, which are not represented in current assessment systems. While progress on Climate Action in Kenya is on track according to the 2023 Sustainable Development Report, major challenges remain, with progress stagnating, in relation to SDGs encompassing poverty, hunger, health and wellbeing [85]. Environmental considerations are important, but while addressing environmental concerns, the social and economic wellbeing of the people who interact with buildings is crucial to the realisation of sustainable design.

While Green Mark Kenya and Green Star SA-Kenya include credits for 'social' criteria such as 'Stakeholder consultation' and 'Inclusive and accessible design' (Green Mark), or 'Local connectivity' (Green Star), social credits only account for more than 25% of the

total assessment in Green Mark. A rating system that truly recognises the contextual dynamics of sustainable design in an African context would include a more equal balance between social, economic and environmental factors, with each constituting approximately 30–35% of the total assessment. What obstacles prevent this? It can be difficult to identify multiple social or economic criteria that are individually verifiable by an external party through a conventional audit trail. This reveals an inherent problem with the design of assessment systems—a structural bias towards quantifiable criteria. A system that properly rewarded social impact—e.g., in local communities—would need to be applied at different stages over a project's lifespan—planning, construction, occupancy—and would rely on qualitative assessment and value judgments by those best placed to monitor local impact, i.e., professionals with familiarity with the local context. Such a rating system would necessarily require a greater degree of monitoring over a longer period of time, increasing the cost.

In conclusion, three fundamental considerations emerge for the evaluation of building assessment systems in developing world contexts:

1. The inclusion of categories identifying local problems, challenges, and objectives.
2. Weighting which criteria to prioritise based on local conditions or dynamics.
3. The inclusion of criteria reflecting an appropriate balance of environmental, social, and economic factors.

Specifically, environmental criteria, such as Energy Efficiency and Indoor Environmental Quality, should take account of the local context, including carbon intensity of energy supply, and adaptation to local climate. Social and economic factors should be developed and regularly updated to reflect local development needs, e.g., based on progress towards Sustainable Development Goals. While locally developed systems (Safari Green Building Index and Green Mark Kenya) begin to address some of these considerations, none of the four assessment systems analysed in this study—when evaluated based on the above criteria—are adequate for the context of Kenya.

**Author Contributions:** Conceptualisation, F.N.C. and R.L.; Methodology, F.N.C.; Formal Analysis, F.N.C. and R.L.; Investigation, F.N.C.; Data Curation, F.N.C.; Writing—Original Draft Preparation, F.N.C. and R.L.; Writing—Review and Editing, R.L.; Visualisation, F.N.C.; Supervision, R.L.; Project Administration, F.N.C. All authors have read and agreed to the published version of the manuscript.

**Funding:** This research received no external funding.

**Institutional Review Board Statement:** The study was conducted according to the guidelines of the Declaration of Helsinki and approved by the Institutional Review Board (or Ethics Committee) of the University of Sheffield (Reference Number 015888, approved 27/07/2017).

**Informed Consent Statement:** Informed consent was obtained from all subjects involved in the study.

**Data Availability Statement:** The original contributions presented in the study are included in the article; further inquiries can be directed to the corresponding author.

**Conflicts of Interest:** The authors declare no conflicts of interest. Informed consent was obtained from all subjects involved in the study.

## Appendix A

| Key | |
| --- | --- |
| Green | Environmental |
| Blue | Social |
| Red | Economic |
| Black | Management/Administrative |

**LEED 4.0 BD+C (New Construction)**

| Category | Weighting Points | (%) | Criteria |
|---|---|---|---|
| Integrative Process | 1 | 1 | |
| Location and Transport | 16 | 14.5 | LEED for Neighbourhood Development location<br>Sensitive land protection<br>High priority site<br>Surrounding density and diverse uses<br>Access to quality transit<br>Bicycle facilities<br>Reduced parking footprint<br>Green vehicles |
| Sustainable Sites | 10 | 9 | Construction activity pollution prevention<br>Site assessment<br>Site development—protect or restore habitat<br>Open space<br>Rainwater management<br>Heat island reduction<br>Light pollution reduction |
| Water Efficiency | 11 | 10 | Outdoor water use reduction<br>Indoor water use reduction<br>Building—level water metering<br>Outdoor water use reduction<br>Cooling tower water use<br>Water metering |
| Energy and Atmosphere | 33 | 30 | Fundamental commissioning and verification<br>Minimum energy performance<br>Building-level energy metering<br>Fundamental refrigeration management<br>Enhanced commissioning<br>Optimise energy performance<br>Advanced energy metering<br>Demand response<br>Renewable energy production<br>Enhanced refrigerant management<br>Green power and carbon offsets |
| Materials and Resources | 13 | 12 | Storage and collection of recyclables<br>Construction and demolition water management planning<br>Building life-cycle impact reduction<br>Building product disclosure and optimisation—environmental product declarations<br>Building product disclosure and optimisation—sourcing of raw materials<br>Building product disclosure and optimisation—material ingredients<br>Construction and demolition waste management |
| Indoor Environmental Quality | 16 | 14.5 | Environmental tobacco smoke control<br>Minimum indoor air quality performance<br>Enhanced indoor air quality strategies<br>Low-emitting materials<br>Construction indoor air quality management plan<br>Indoor air quality assessment<br>Thermal comfort<br>Interior lighting<br>Daylight<br>Quality views<br>Acoustic performance |

**LEED 4.0 BD+C (New Construction)**

| Category | Weighting Points | (%) | Criteria |
|---|---|---|---|
| Innovation | 6 | 5 | Innovation<br>LEED Accredited Professional |
| Regional Priority | 4 | 3 | |

**Green Star SA-Kenya**

| Category | Weighting (%) | Criteria |
|---|---|---|
| Management | 10 | Green Star KE Accredited Professional<br>Commissioning clauses<br>Building turning<br>Independent commissioning agent<br>Building User's Guide<br>Environmental management<br>Waste management |
| Indoor Environmental Quality | 15 | Ventilation rates<br>Air change effectiveness<br>Carbon dioxide monitoring and control<br>Daylight<br>Daylight glare control<br>High frequency ballasts<br>Electric lighting levels<br>External views<br>Thermal comfort<br>Individual comfort control<br>Hazardous materials<br>Internal noise levels<br>Volatile organic compounds<br>Formaldehyde minimisation<br>Mould prevention<br>Tenant exhaust riser<br>Environmental tobacco smoke (ETS) avoidance energy |
| Energy | 25 | Energy—conditional requirement<br>Greenhouse gas emissions<br>Energy sub-metering<br>Lighting power density<br>Lighting zoning<br>Peak energy demand reduction |
| Transport | 9 | Provision of car parking<br>Fuel efficient transport<br>Cyclist facilities<br>Commuting mass transport<br>Local connectivity |
| Water | 14 | Occupant amenity water<br>Water meters<br>Landscape irrigation<br>Heat rejection water<br>Fire system water consumption |

**Green Star SA-Kenya**

| Category | Weighting (%) | Criteria |
|---|---|---|
| Materials | 13 | Recycling waste storage<br>Building reuse<br>Shell and core or integrated fit-out<br>Concrete/steel<br>PVC minimisation<br>Sustainable timber<br>Design for disassembly<br>Dematerialisation<br>Local sourcing |
| Land Use and Ecology | 7 | Ecology—conditional requirement<br>Topsoil<br>Reuse of land<br>Reclaimed contaminated land<br>Change of ecological value |
| Emissions | 7 | Refrigerant / gaseous ODP<br>Refrigerant GWP<br>Refrigerant leaks<br>Insulant ODP<br>Watercourse pollution<br>Discharge to sewer<br>Light pollution<br>Legionella<br>Boiler and generator emissions |
| Innovation | - | Innovative strategies and technologies<br>Exceeding Green Star KE benchmarks<br>Environmental design initiatives |

**Green Mark Kenya**

| Category | Weighting (%) | Criteria |
|---|---|---|
| Sustainable Site, Planning Development and Management | 15 | Life cycle cost and service planning<br>*Site Planning and Development*<br>Site selection<br>Building development density<br>Building orientation and form<br>Maximising usage of built and green spaces<br>Protect or restore habitat<br>Reduce heat island effect<br>Erosion control and landscape management<br>Responsible construction<br>Light pollution<br>*Social Value Maximisation*<br>Inclusive and accessible design<br>Emergency response<br>*Water Ecosystem Management*<br>Storm water design and management<br>Integrated pest management<br>*Sustainable Transport*<br>Non-motorised transport<br>Mass, efficient and low-emitting vehicular transport |

**Green Mark Kenya**

| Category | Weighting (%) | Criteria |
|---|---|---|
| Water Efficiency | 10 | Water use management<br>Water harvesting<br>Water efficient fixtures and fittings<br>Grey water recycling<br>Water efficient landscaping and irrigation systems<br>Water quality<br>Life cycle cost and service planning<br>Site planning and development<br>Social value maximisation<br>Water ecosystem management<br>Sustainable transport |
| Energy Efficiency | 20 | Optimise energy performance<br>Commissioning and re-commissioning of building energy systems<br>Energy efficient equipment, appliances, fittings<br>Energy monitoring<br>Renewable energy<br>Light zoning |
| Sustainable Materials and Technology | 10 | *Environmentally Responsible Materials*<br>Rapidly renewable materials<br>Low-emitting materials<br>Locally sourced materials and products<br>Responsible sourcing of materials<br>*Resource Efficient and Technology*<br>Reused and recycled materials<br>Appropriate building technology<br>Construction waste management |
| Indoor Environmental Quality | 30 | *Ventilation and Thermal Comfort*<br>Climate responsive design<br>Natural ventilation, heating and cooling<br>User-friendly heating and ventilation systems<br>Air change effectiveness<br>*Light and Visual Comfort*<br>Natural lighting<br>User-friendly lighting systems<br>Glare control and view out<br>Efficient artificial lighting fittings<br>*Indoor Air Quality*<br>Minimum IAQ performance<br>Tobacco smoke control<br>Indoor air quality testing and monitoring<br>*IndoorAir Pollution*<br>Mould prevention<br>*Acoustic Comfort*<br>Internal noise level |
| Maintenance and Management | 10 | *Community Engagement*<br>Stakeholder consultation<br>Building user manual<br>Post occupancy evaluation<br>Green procurement policy<br>*Site Waste Management*<br>Operation waste management<br>Building exterior management |
| Innovation | 5 | - |

**Safari Green Building Index**

| Category | Weighting (%) | Criteria |
|---|---|---|
| Prerequisite Requirements | - | Environmental laws (NEMA)<br>Space and occupation<br>Building development or distances and zoning<br>Site boundary<br>Universal<br>Accessibility<br>Building user manual / building information |
| Strategies General Description/Recommendation | 5 | Sustainable site planning<br>Landscaping and irrigation |
| Passive Design Strategies | 45 | *Solar Control*<br>Building orientation<br>Space allocation within the building<br>Building form and shape<br>Openings<br>Natural lighting<br>Sun shading / solar control<br>Thermal mass<br>Passive heating or passive cooling<br>Building finishes<br>*Natural Ventilation*<br>Natural ventilation and cooling |
| Energy Efficiency | 10 | Energy efficient equipment/appliances/fittings<br>Renewable energy |
| Resource Efficiency | 30 | *Materials*<br>Choice of building material (external)<br>Choice of building materials (internal)<br>*Water Supply and Drainage*<br>Water supply<br>Storm water drainage<br>*Waste Management*<br>Solid waste management<br>Waste water management |
| Noise Control and Acoustic Design | 5 | Noise control and acoustics |
| Innovation | 5 | Sustainable design innovation |

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
