# Peer review of "Always with the Best Intentions? Interrogating the Use of Sustainable Building Assessment Systems in Developing Countries: Kenya"

_sustainability, doi:10.3390/su16093868_

Round 1

Reviewer 1 Report

Comments and Suggestions for Authors

The article discusses the applicability of international assessment systems (such as LEED-US and Green Star SA-Kenya) and local development systems (such as Green Mark Kenya and Safari Green Building Index) in the context of Kenya. It compares the relative weight allocation of these systems across different categories, highlighting key considerations when implementing these systems in developing countries.

However, the article still has room for improvement in several aspects:

1. Logical Flow: The second section, "Green or Sustainable," appears redundant and could be merged with the introductory section. Additionally, the analysis related to Table 4 lacks a strong connection to Kenya.

2. Content Analysis: In the third section, the statement "However, compared to developed countries, developing nations prioritize economic and social issues" lacks supporting evidence. Table 1 also lacks in-depth analysis. When discussing the application of international systems in Kenya, the article lacks case studies on the challenges and limitations faced during practical implementation. Similarly, when discussing localized assessment systems, there is a dearth of information on how these systems better integrate social and economic factors and their effectiveness and challenges in real-world applications.

3. Conclusion Summary: The article does not provide a clear conclusion with specific recommendations for improving assessment systems, especially regarding enhancing flexibility and adaptability to meet the diverse needs of different countries.

4. Formatting Issues: There are punctuation errors at the end of the first paragraph in the fourth section, as well as an error in punctuation at line 308. Additionally, lines 325-328 in the sixth section have formatting issues.

Comments on the Quality of English Language

There are punctuation errors at the end of the first paragraph in the fourth section, as well as an error in punctuation at line 308. Additionally, lines 325-328 in the sixth section have formatting issues.

Reviewer 2 Report

Comments and Suggestions for Authors

In general

Developing and improving a green building rating system for Kenya is an important and pressing issue. Finding a balance between passive and mechanical ventilation, as well as between passive and mechanical cooling and heating systems, is an important task for the architect.

However, in this article there is no causal relationship between the conclusions section and other sections. The Conclusions section contains information that was not discussed in detail elsewhere in the article.

LEED rating systems are poorly described. This greatly reduces the strength of the argument for the benefits of the new green building rating system over LEED.

This article barely reviews the literature from the past three years. In addition, literary sources are cited in the text of the article, but are not contained in the reference section.

In specific

If the title of the article contains the term “Sustainable Building Assessment Systems”, then an explanation of this term should be contained in the first part of the introduction.

 Lines 29-36. What is measured in BREEAM and LEED systems: environment or sustainability?

Cole 2018 is not listed in the reference section.

Line 38. LEED was launched in 1998

Lines 156-157. It was written that “The climates of Nairobi and Mombasa in Kenya, for example, favour passive design strategies over mechanical ventilation”.

Please indicate a literary source.

Lines 170-173.It was written that “While LEED is marketed as representing ‘a transformative milestone in the built environment’s alignment with a low-carbon future’ (USGBC, 2024b), evidence for improvement in energy performance of LEED-certified buildings remains inconclusive (Newsham et al., 2009; Amir et al., 2019)”.

More detailed evidence needs to be provided.

Newsham et al., 2009 refers to an analysis of the first ten years of a LEED-certified practice.

Amir et al., 2019 is not listed in the reference section.

It should be noted that there is a significant difference between LEED v2.2 and LEED v4.

Please indicate a critical analysis of LEED v4.

Line 268. Where is Appendix 1?

Table 3. What LEED rating system 1 presented in Table 3?

LEED contains more than 30 rating systems (e.g., LEED New Construction, (LEED-NC), Commercial Interiors (LEED-CI), and etc. These systems differ significantly from each other.

A problem arises when comparing Tables 2 and 4. For example, in the LEED system, Table LEED 2 shows the four certification levels when the overall LEED score is 120, and Table 4 shows the highest possible % for the seven categories when the overall LEED score is 100.

Lines 322-323. Where is it mentioned that renewable energy sources in the US make up approximately 17% of the energy mix?

Lines 323-324. It was written “It is no surprise therefore that energy is emphasised in the rating systems originating in these other countries”. Please describe the energy focus in South Africa.

Lines 325-328. It was written “Nairobi (Kenya) Mombasa (Kenya) Johannesburg (South Africa) Cfb As. Cwb. Subtropical highland Tropical savanna Subtropical highland”

It is unclear.

Lines 340-341. Figure 2 is presented in an implicit manner. The top panel shows three cities, and the bottom panel shows three geographic locations.  How does this compare? This figure must be removed.

Lines 397-404 and Table 5. Table 5 shows two LEED commercial interiors (LEED-CI) v3 gold-certified projects and one LEED Core and Shell (LEED-C-and-S) v3 gold-certified project. Therefore, only the first two LEED-certified projects can be compared.  The third project cannot be compared with the first two projects. The third project relates to a different rating system and a different level of certification.  There are several studies in the literature that have analyzed such LEED-certified projects.  The materials and resources (MR) and indoor environmental quality (EQ) categories of LEED Gold-certified projects showed low achievement in both the US and non-US countries. This is not a problem specific to the Kenyan construction sector.

Lines 406-434.  The cost issue in LEED-certified projects has been described at the quality level rather than the quantity level.

Lines 459-462.  It was written that “This holds greater significance for developing nations… plays a notably more crucial role in these countries”.

If the authors used literary sources more than twenty years ago to indicate the importance of the problem, then this sounds unconvincing.

Lines 463-485.  This subsection contains only conclusions about the advantages of passive heating and cooling over mechanical heating. Where is the analysis of the literature on comfort in terms of quantitative assessment?

Lines 517-521.  It was written “Assessment is a service that is sold to customers expecting a desirable outcome…” Such a conclusion must be based on both quantitative assessments and causal phenomena.

Lines 534-537. It was written that “These models may be appropriate for Western countries, however, for countries like Kenya, whose energy largely originates from sustainable sources, the focus on energy as a main performance indicator is not appropriate”.

The reviewer suggests that the following phrase, “for countries like Kenya, whose energy largely originates from sustainable sources,” is important in this manuscript.

Therefore, the literature review section should contain the components of the term “sustainable sources” and a detailed explanation of how this fact influences green building strategy. The literature review section should also include examples of successful and unsuccessful applications of LEED in developing countries.

Lines 563-564. It was written that “Finally, a general oversight in sustainable building assessment systems is that buildings are assessed against predicted performance as opposed to actual performance.”

However, the manuscript does not provide a comparative analysis of buildings certified under green rating systems in terms of “predicted performance” versus “actual performance issues.”

Reviewer 3 Report

Comments and Suggestions for Authors

The paper titled "Always with the Best Intentions? Interrogating the Use of Sustainable Building Assessment Systems in Developing Countries: Kenya", fills a knowledge gap by comparing local Kenyan sustainable building rating method with worldwide methods. It focusses on integrating social and economic aspects with the environment to provide a more comprehensive approach toward sustainability. The provision of a local case study adds to the study importance. 

The paper can be improved by considering the following points: 

1. Incorporating more recent research studies from both developed and developing countries.  

2. Develop the conclusion by linking it back to the research objectives and adding a concluding remark. 

3. Eliminating self-plagiarism, which was found throughout the paper, taken from the PhD thesis titled "SITUATING THE DISCOURSE OF SUSTAINABLE DESIGN IN NAIROBI, KENYA". This can be seen from the iThenticate report (36% match).  

Reviewer 4 Report

Comments and Suggestions for Authors

The article aimed to assess the reliability of International standards such as BREEAM and LEED. The article claims that international rating systems such as LEED often focus on a Western construct of sustainability featuring a systematic bias towards global rather than local perspectives.

Major revisions in the text and minor language editing are required.

Abstract:

“In pursuit of objectivity, the measurement of acontextual parameters (e.g., energy performance) is prioritized at the expense of those contingent on local circumstances or climate.”

It is not clear what “acontextual parameters” means. Could you please explain it briefly?

Introduction:

Line 41: “It is estimated that there are now over 600 green building rating systems across the globe (Doan et al., 2017).” The reference used to support the statement is from 2017. Rephrase that sentence or update the reference.

Figure 1. Could you please explain what “A” and “B” mean?

The Methods section is clear. However, there is room for improvement:

Lines 324 to 327: It is difficult to understand the text. I suggest the authors use a table to display the information.

Conclusions:

Lines 499-501. The conclusion section is not the right place to include references.

“Methodologies based on a process of rational scientific evolution would have responded to the growing scientific consensus around this issue, known as ‘the performance gap’ (Menezes et al.,2012).”

The first paragraph (from line 487 to line 521) should be moved to the introduction section. It helps identify the problem, but it is not a conclusion.

Lines 550-557 are the actual conclusions of the study:

“In conclusion, three fundamental considerations emerge for the evaluation of building assessment systems in developing world contexts:

1. The inclusion of categories identifying local problems, challenges, and objectives.

2. Weighting which prioritises criteria based on local conditions or dynamics.

3. The inclusion of criteria reflecting an appropriate balance of environmental, social, and economic factors.”

Could you be more specific about local problems and challenges, priorities based on local conditions, and social factors in Kenya?

There is relevant data in lines 450 to 455 that could be used to draw meaningful conclusions.

Use the information from the Methods and Discussion sections to Show conclusions based on data. The current conclusions are too general and do not help readers understand the depth of the analysis.

Comments on the Quality of English Language

Minor editing of English language required

Round 2

Reviewer 1 Report

Comments and Suggestions for Authors

The paper discusses the suitability and impact of using Sustainable Building Assessment Systems (SBAS) in developing countries, especially in Kenya, which is of great significance for promoting global sustainable development.

After the author’s revisions and improvements, the paper presents a clear logical structure, employs relatively rigorous methods, and reaches conclusions that are broadly applicable.

Author Response

Response to reviewers
Please find attached a revised manuscript for the above submission, with revisions highlighted in yellow.

Kind thanks to the reviewers for their additional comments. The manuscript has been revised accordingly. For clarity, we have included all of the reviewers’ comments below in black, with our responses outlined in red.
Review 1
Comments and Suggestions for Authors
After the author’s revisions and improvements, the paper presents a clear logical structure, employs relatively rigorous methods, and reaches conclusions that are broadly applicable.

Thank you for your comments – they are much appreciated.

Reviewer 2 Report

Comments and Suggestions for Authors

Review of “Always with the Best Intentions? Interrogating the Use of Sustainable Building Assessment Systems in Developing 3 Countries: Kenya”

In general

The authors have significantly improved the scientific quality of the manuscript. However, Table 5 contains an incorrect comparison. Review articles rather than original articles were used to interpret the results of Table 5. To interpret the results of Table 5, the authors interpreted the original articles by citing review articles. Such citation may lead to unintentional misrepresentation of facts.

This manuscript may be accepted after significant revisions.

In specific

In the current study, Table 5 shows three LEED-certified projects in Nairobi, Kenya. However, two out of three LEED-certified projects are in the same rating system (i.e., Commercial Interiors) and at the same certification level (i.e., Gold). In contrast, one out of three LEED-certified projects is in a different rating system (i.e., Core and Shell) and at a different level of certification (i.e., Certified). It should be noted that different LEED rating systems and different certification levels have different LEED certification strategies. Therefore, these three LEED-certified projects in Table 5 cannot be compared under any circumstances.

The reviewer suggested comparing two LEED-CI v3 gold-certified projects from Nairobi, Kenya with Pushkar's study (see Pushkar, S. Evaluating LEED commercial interior (LEED-CI) projects under the LEED transition from v3 to v4: The differences between China and the US. Heliyon 2020, 6, e04701.)

The reviewer's table shows that Nairobi, Kenya, China and the US have similar achievements in the Materials and Resources and Indoor Environmental Quality categories.

Reviewer's table. Leadership in Energy and Environmental Design Commercial Interiors (LEED-CI) version 3 (v3) 2009 gold-certified projects

LEED v3 2009 category

Maximum possible points

The current study

Pushkar [2020]

Nairobi, Kenya, achieved points

Median and 25th–75th centiles, achieved points

Citibank Gigiri Branch

World Bank Group

China

The USA

Materials and Resources (MRs)

14

0.0

5.0

4.0 4.0–6.0

4.5 3.0–6.0

Indoor Environmental Quality (EQ)

17

8.0

7.0

8.5 7.0–10.0

9.0 8.0–10.0

Lines 474-481.

The authors used two review articles (Geng et al., 2019 and Afroz, Z. et al., 2020) to argue that indoor environmental quality in LEED-certified buildings has challenges in terms of the user's individual control of their environment. I would like to get information from original articles, and not from a review article.

When the authors discussed the issue of auditing and the cost of building materials in LEED-certified projects, a review article by Ascione et al., (2022) was used. However, Ascione et al. (2022) cited the Wu et al., (2017) study “In the office sector, there is a point chasing in SS [Sustainable Sites] and IEQ Indoor Environmental Quality] credits and a point allocation problem in MR [Materials and Resources credits]”.  Information on the issue of auditing and the cost of building materials in LEED-certified projects would be desirable to obtain from the original article, and not from a review article.

Line 466. There is a misspelling “Na[i]robi”.

Author Response

Response to reviewers

Please find attached a revised manuscript for the above submission, with revisions highlighted in yellow.
Kind thanks to the reviewers for their additional comments. The manuscript has been revised accordingly. For clarity, we have included all of the reviewers’ comments below in black, with our responses outlined in red.

Review 2
Comments and Suggestions for Authors
In general
The authors have significantly improved the scientific quality of the manuscript. However,Table 5 contains an incorrect comparison. Review articles rather than original articles were used to interpret the results of Table 5. To interpret the results of Table 5, the authors interpreted the original articles by citing review articles. Such citation may lead to unintentional misrepresentation of facts.
Thank you for this observation. Original articles have now been cited.

This manuscript may be accepted after significant revisions.
In specific
In the current study, Table 5 shows three LEED-certified projects in Nairobi, Kenya. However, two out of three LEED-certified projects are in the same rating system (i.e., Commercial Interiors) and at the same certification level (i.e., Gold). In contrast, one out of three LEED-certified projects is in a different rating system (i.e., Core and Shell) and at a different level of certification (i.e., Certified). It should be noted that different LEED rating systems and different certification levels have different LEED certification strategies.
Therefore, these three LEED-certified projects in Table 5 cannot be compared under any circumstances.

Table 5 has been modified to include the two ID+C Gold-certified buildings only, and a new table (Table 6) has been inserted to include the other BD+C-certified building.

The reviewer suggested comparing two LEED-CI v3 gold-certified projects from Nairobi, Kenya with Pushkar's study (see Pushkar, S. Evaluating LEED commercial interior (LEED-CI) projects under the LEED transition from v3 to v4: The differences between China and the US. Heliyon 2020, 6, e04701.)

The reviewer's table shows that Nairobi, Kenya, China and the US have similar
achievements in the Materials and Resources and Indoor Environmental Quality categories.

Thank you for this suggestion. This reference and the data have been incorporated into the analysis:

These results are similar to a study examining LEED (ID+C 3.0) certified buildings in China and the U.S., which found that the median number of points attained for Indoor Environmental Quality was 50% in China and 53% in the U.S.; and 29% in China and 32% in the U.S. for Materials & Resources (Pushkar, 2020). Preliminary analysis suggests, however, that these two categories should be easier to achieve within a Kenyan context.

Lines 474-481.
The authors used two review articles (Geng et al., 2019 and Afroz, Z. et al., 2020) to argue that indoor environmental quality in LEED-certified buildings has challenges in terms of the user's individual control of their environment. I would like to get information from original articles, and not from a review article.

These references have been replaced with original articles:

Altomonte, S., Schiavon, S., Kent, M. G., & Brager, G. (2019). Indoor environmental quality and occupant satisfaction in green-certified buildings. Building Research & Information 47 (3), pp. 255-274.

He, Y., Wong, N. H., Kvan, T., Liu, M., & Tong, S. (2022). How green building rating systems affect indoor thermal comfort environments design. Building and Environment 224, 109514.

When the authors discussed the issue of auditing and the cost of building materials in LEED-certified projects, a review article by Ascione et al., (2022) was used. However, Ascione et al. (2022) cited the Wu et al., (2017) study “In the office sector, there is a point chasing in SS [Sustainable Sites] and IEQ Indoor Environmental Quality] credits and a point allocation problem in MR [Materials and Resources credits]”. Information on the issue of auditing and the cost of building materials in LEED-certified projects would be desirable to obtain from the original article, and not from a review article

Thank you for this suggestion. We have replaced the reference as follows:

Wu, P., Song, Y, Shou, W, Chi, H., Chong, H. & Sutrisna, M. (2017). A comprehensive analysis of the credits obtained by LEED 2009 certified green buildings. Renewable and Sustainable Energy Reviews 68 (1), pp. 370-379.

Line 466. There is a misspelling “Na[i]robi”.

Thank you for spotting this – it has been corrected.

Reviewer 3 Report

Comments and Suggestions for Authors

The authors have improved the quality of the article; all the best with their future research work.

With respect to the self-plagiarism part, I will leave this to the editor to make a decision about it.

Comments on the Quality of English Language

Minor grammatical mistakes should be corrected, for instance: 

1) "A rating system that truly recognised the contextual dynamics of", should be changed to "A rating system that truly recognizes the contextual"

 2) "What obstacles prevent this? It is difficult to identify multiple social or", should be changed to " What obstacles prevent this? It can be difficult to identify multiple social or"

Author Response

Response to reviewers
Please find attached a revised manuscript for the above submission, with revisions highlighted in yellow.
Kind thanks to the reviewers for their additional comments. The manuscript has been revised accordingly. For clarity, we have included all of the reviewers’ comments below in black, with our responses outlined in red

Review 3
Comments and Suggestions for Authors
The authors have improved the quality of the article; all the best with their future research work. With respect to the self-plagiarism part, I will leave this to the editor to make a decision about it.

Comments on the Quality of English Language

Minor grammatical mistakes should be corrected, for instance:
1) "A rating system that truly recognised the contextual dynamics of", should be changed to "A rating system that truly recognizes the contextual"
2) "What obstacles prevent this? It is difficult to identify multiple social or", should be changed to " What obstacles prevent this? It can be difficult to identify multiple social or"

Thank you for your comments – they are much appreciated. These errors have now been corrected.

Reviewer 4 Report

Comments and Suggestions for Authors

THE AUTHORS HAVE ADDRESSED THE PREVIOUS COMMENTS.

Author Response

Response to reviewers

Please find attached a revised manuscript for the above submission, with revisions highlighted in yellow.

Kind thanks to the reviewers for their additional comments. The manuscript has been revised accordingly. For clarity, we have included all of the reviewers’ comments below in black, with our responses outlined in red

Review 4
Comments and Suggestions for Authors

The authors have addressed the previous comments.

Thank you for your comments – much appreciated.

Round 3

Reviewer 2 Report

Comments and Suggestions for Authors

Accept